# Cell Salvage in Oncological Surgery, Peripartum Haemorrhage and Trauma

**Lidia Mora Miquel** [1,*], **Susana Manrique Muñoz** [2] **and Marc Maegele** [3]

1 Department of Anaesthesiology, Intensive Care and Pain Clinic, Vall d'Hebron Trauma, Rehabilitation and Burns Hospital, Autonomous University of Barcelona, 08035 Barcelona, Spain

2 Department of Anaesthesiology, Intensive Care and Pain Clinic, Vall d'Hebron Mother and Child Hospital, Autonomous University of Barcelona, 08035 Barcelona, Spain; smanrique@vhebron.net

3 Department for Trauma and Orthopaedic Surgery, Cologne–Merheim Medical Centre, Institute for Research in Operative Medicine, University Witten/Herdecke, 51109 Cologne, Germany; marc.maegele@t-online.de

\* Correspondence: 33169lmm@comb.cat

**Abstract:** Oncological surgery, obstetric haemorrhage and severe trauma are the most challenging conditions for establishing clinical recommendations for the use of cell salvage. When the likelihood of allogeneic transfusion is high, the intraoperative use of this blood-saving technique would be justified, but specific patient selection criteria are needed. The main concerns in the case of oncological surgery are the reinfusion of tumour cells, thereby increasing the risk of metastasis. This threat could be minimized, which may help to rationalize its indication. In severe peripartum haemorrhage, cell salvage has not proven cost-effective, damage control techniques have been developed, and, given the risk of fetomaternal alloimmunization and amniotic fluid embolism, it is increasingly out of use. In trauma, bleeding may originate from multiple sites, coagulopathy may develop, and it should be evaluated whether re-transfusion of autologous blood collected from uncontaminated organ cavities would be feasible. General safety measures include washing recovered blood and its passage through leukocyte depletion filters. To date, no well-defined indications for cell salvage have been established for these pathologies, but with accurate case selection and selective implementation, it could become safe and effective. Randomized clinical trials are urgently needed.

**Keywords:** cell salvage; peripartum haemorrhage; oncological surgery; trauma bleeding; autologous blood re-transfusion

## 1. Introduction

Restrictive allogeneic blood transfusion has been recommended as best practice in most surgical procedures. The use of intraoperative cell salvage (ICS) shows great variability in clinical practice, as some pathologies may require specific patient-centred decision making, and it remains difficult to standardise an indication. Its value for minimizing perioperative allogeneic blood transfusion has been widely acknowledged, as in the context of orthopaedic or cardiac surgery [1]. High-quality studies are still lacking that manage to avoid patient selection and treatment biases, as some results in previous studies have favoured the use of ICS.

The incorporation of ICS in very specific pathologies should be evaluated. A narrative review on the use of ICS in oncological surgery, peripartum haemorrhage (PPH) and trauma is presented based upon a selective review of the literature accessible through PubMed and Google Scholar using "cell salvage", "autologous blood re-transfusion", "oncological surgery", and "peripartum haemorrhage "trauma bleeding" as key words. The bibliographic search was also supported by EMBASE and Web of Science.

The aim of this review focuses on the technical implications and the difficulty of clinical decision making for the use of ICS in highly dynamic and complex clinical scenarios, considering its limitations and contraindications.

The recovery of autologous blood in oncological surgery, PPH and major traumatic bleeding has as main drawbacks the potential risk of re-transfusion of cellular or humoral elements that are considered harmful. The primary concerns are the potential return of septic, malignant or embolic agents, the risk of fetomaternal alloimmunization, the requirement to have a non-contaminated organic cavity in case of trauma and the dynamic and unexpected condition of the bleeding.

The presence of contaminants in recovered blood should be considered as a contraindication (Table 1), but this is controversial with little evidence to support widespread use in such conditions. Filtering and complete washing of red blood cells (RBC) with 0.9% saline [2–4] are considered indispensable premises to avoid the incorporation of potentially harmful elements.

**Table 1.** Substances Not Suitable for Reinfusion.

| | |
|---|---|
| Topical coagulants, drugs | Activated leukocytes and platelets |
| Methyl methacrylate (bone cement) | Bacteria, endotoxins |
| Irrigation solutions (iodized) | Enzymes from cellular disruption |
| Topical antibiotics | Activated complement: C3a, C5a |
| Fat cells, malignant cells, mesothelial cells | Activated fibrinolytic products, plasmin |
| Bone splinters | Fibrin, split products, and D-dimers |
| Smoke from electrocautery (carbon oxide) | Organic fluids: faecal, urine, ascites, gastric acid, bile, amniotic fluid |

The use of ICS ought to be well-protocolized to optimize its benefits while minimizing risks and potential complications, and it should also be cost-effective [5–7].

## 2. Current Recommendations and Guidelines

The European Guidelines on severe bleeding provide suggestions on the use of ICS in obstetrics and oncological surgery, with grades of evidence ranging from 2B to C. In the European Guidelines on Traumatic Bleeding, ICS is not yet considered, but this topic will be reviewed for the next editions in both Guidelines [8,9]. Ideally, ICS should be part of a comprehensive patient blood management (PBM) program, as outlined in the UK NICE Guidelines (www.nice.org.uk/guidance/ng24; www.transfusionguidelines.org, accessed on 9 February 2022), as well as by The National Blood Authority in Australia (https://www.blood.gov.au/pbm-guidelines, accessed on 9 February 2022), and be framed within a hemovigilance program (www.shotuk.org/resources/current-resources/data-drawers/cell-salvage-cs-data-drawer/, accessed on 9 February 2022), which also highlights the multidisciplinary involvement of blood banks and transfusion departments.

## 3. Criteria Determining RBC Volume to Re-Transfuse and Technical Aspects

Patients should be identified preoperatively with potential intraoperative blood loss of at least 500–1000 mL or between 10% [5] and 20% [8] of estimated blood volume and who could maximize the percentage of autologous blood used [10,11], with the recovery of 1–2 RBC units expected [12,13].

The RBC volume to be re-transfused varies according to blood loss, haematocrit, and the capacity of the processing system. The ideal blood product would yield the following:

(i.) Viable and efficient RBC recovery of 90%;
(ii.) Washout of 90%;
(iii.) Haematocrit 55–80%;
(iv.) Free haemoglobin clearance of 95%;
(v.) Albumin clearance of 96%.

The centrifugation process should separate RBCs and remove detritus, like free haemoglobin (Hb), plasma, cytokines, fibrin, complement fragments, platelets, leuko-

cytes, and heparin. The technique involves heparinisation of the serum that irrigates the surgical field, using sodium heparin 25,000–30,000 IU in 1000 mL saline 0.9% or citrate dextrose 60–80 drops/min, or even argatroban [14]. Blood damage is reduced when blood is diluted in saline while being suctioned from the surgical field. There is a dynamic saline-air interface in the most superficial areas that could affect the efficiency of erythrocyte recovery, especially if the suction pressure is high. It is recommended that the vacuum pressure oscillates between −100 and −150 mmHg and does not exceed −300 mmHg to avoid superficial aspiration due to turbulence [15,16]. The importance of double suction in obstetrics to differentiate blood suction from amniotic fluid suction has been highlighted. In case of massive haemorrhage, the need for high-speed suction could increase the pressure but could also increase the risk of cell lysis.

As for the diameter of the filter pores for the re-transfusion of recovered blood, 40 to 100 microns have been traditionally recommended. A recent study [17] has evaluated the efficacy of cell washing and recovery by measuring complete blood count, free haemoglobin, complement factors and D-dimer in both concentrated and filtered blood. The best device achieved a significantly higher reduction of contaminants. High-precision nanotechnology was used to achieve pores of exactly 2.3 microns in diameter, allowing only fluids and solutes to pass through, while RBCs remained on top of the filtration layer [17]. A future direction of research will be to evolve the quality of blood filters to ensure re-transfusion of only those elements that are beneficial.

## 4. Safety and Quality Considerations of ICS

Intraoperative cell salvage is a costly procedure and could be inefficient if not well protocolized. Emergency room and operating room teams should be able to communicate in short loops and conduct short debriefing sessions [18] to avoid delays and interference with other medical orders or procedures. Incidents and complications need to be recorded, and quality indicators must be defined and monitored for improvement.

Different designs of ICS devices allow for different blood processing, and they can operate continuously or discontinuously. Various dynamic recovery sizes are available to suit the speed and magnitude of the bleed. There are also variances in centrifugation speed, washing method, haematocrit sensitivity and priming [6,7,19–22]. The selection of the device shall be made according to the needs by evaluating the differential characteristics of each application, as outlined in Figure 1.

Bacterial contamination of recovered blood represents a challenge. Some studies have assessed the efficacy of washing and filtering for decontamination. An experimental ex vivo nested cohort study [23] was conducted to prove the efficacy of washing combined with leukocyte depletion filters (LDF). Bacterial concentrations were reduced by 85.2%, 91.5%, and 93.9% for *E coli*, *S pseudintermedius*, and *P aeruginosa*, respectively, after washing ($p < 0.0001$), and bacterial concentrations were reduced by 99.9%, 100%, and 100% respectively, after the first filtration ($p < 0.0001$). After the second filtration, no bacteria could be isolated (100% reduction). Generally, it is not recommended to reinfuse blood after 6 h of processing to avoid contamination.

The risk of haemolysis increases with the presence of free radicals and a concentration of free Hb >5 mol/L. Pressurizing blood for reinfusion at a higher rate is not indicated [15,16,24,25]. There is a potential risk of air or amniotic fluid embolism, dilutional coagulopathy, protein loss, septic contamination and disseminated infection, transfusion of activated platelets and clotting factors and activated polymorphonuclear leukocytes. This could lead to a general impairment of coagulation. Re-transfusion of more than 15 units of autologous blood has not been recommended [26]. There is also the risk of microembolization and of an increase in circulating inflammatory markers [27]. The possibility of kidney failure [28,29] was more frequently observed when systems with no washing were used. The potential reinfusion of mesothelial cells [30] has also raised the concern of a recurrence of neoplasms and metastatic spread. Hypotensive episodes have also been reported due to high-velocity blood flow through LDF [31,32].

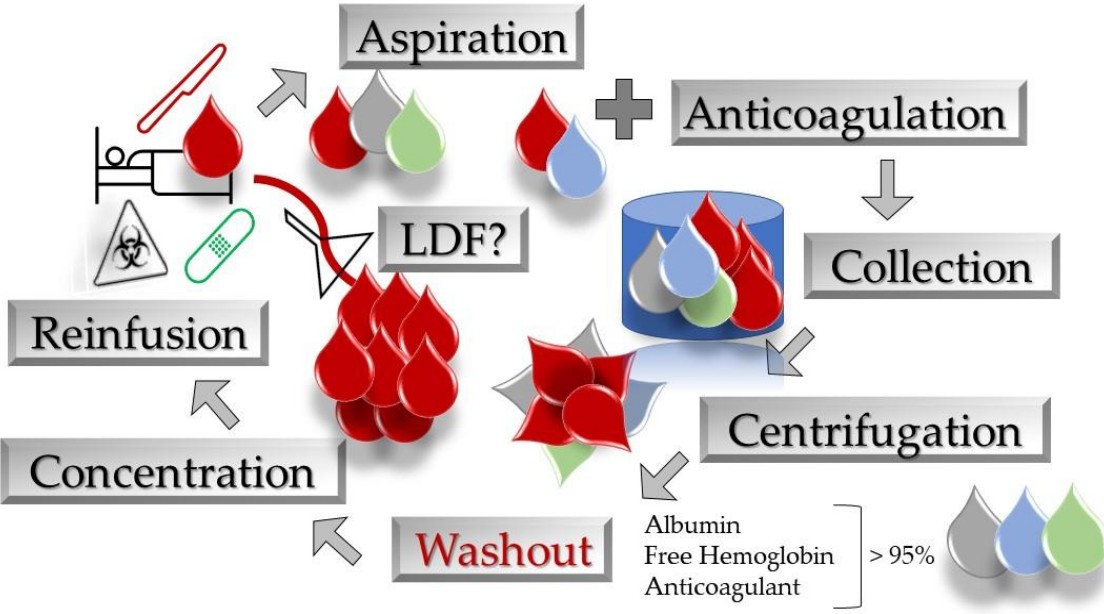

**Figure 1.** Conceptual workflow of intraoperative cell salvage (ICS). LDF: leukocyte depletion filters.

In the context of a severe haemorrhage, there is fear of causing or worsening coagulation function, so the possibility of differentiating, analysing, and individualizing unexpected but activated cellular components may help to discriminate at which point it would not be indicated to reinfuse blood in patients at risk of absolute coagulation decontrol, as in the case of major PPH or severe acute coagulopathy of trauma. Therefore, flow cytometry, immunofluorescence and viscoelastic coagulation measurement techniques could support better orientation [30,32,33].

## 5. Cell Salvage in Oncological Surgery

In cancer procedures, there is a common concern about the risk of possible reinfusion of malignant cells, which could theoretically cause dissemination and favour metastasis formation. However, there is no absolute contraindication to the use of ICS in oncological surgery. The main benefit could translate into reducing the need for allogeneic transfusion, hence reducing transfusion-related immunomodulation (TRIM) [34] with no apparent risk of decreased long-term survival from an oncological perspective. Given that allogeneic transfusion has been associated with increased cancer recurrence through immunomodulation, autologous salvaged blood would theoretically be preferable to allogeneic transfusion.

### 5.1. Tumour Cells in Salvaged Blood Considered with Non-Metastatic Potential

The correct understanding and comprehension of the metastatic risk of a tumour would be the answer to alleviate concerns about the use of ICS in cancer surgery [35], but a source of controversy remains regarding the systemic dissemination of reinfused tumour cells without knowing precisely what their function will be. Studies in cancer surgery have attempted to demonstrate the metastatic ineffectiveness of reinfused tumour cells [35]. Ideally, the recovered blood should be free of tumour cells or contain a significantly lower count compared to the patient's original circulating load. Recent literature has provided information on tumour recurrence, progression, and overall survival rates [35]. In addition, tumour cells in the recovered blood seem to lose the ability to replicate [35]. It is not possible to ensure the absolute metastatic ineffectiveness of tumour cells recovered during

surgery, but based on the evidence, it can be estimated that they would not significantly alter the prognosis and thus could lead to more permissiveness in the use of ICS.

Previous theories have claimed that the re-administration of malignant cells from ICS blood is counterproductive for the patient. Nevertheless, it was estimated in some studies that 0.01–0.000001% of circulating malignant cells could have the potential to form metastatic lesions [36,37]. Cell salvage could be cost-effective if substantial blood loss is expected in major oncological surgeries and defined selection criteria are established [38]. In a meta-analysis of 10 studies, the likelihood for cancer recurrence or development of metastases with ICS for cancer surgery was not increased (OR 0.65; 95% CI 0.43–0.98; $p$ = 0.039). Although none of the included studies were randomized trials, there was neither evidence for publication bias nor a high degree of between-study variability [39]. Irradiation of the salvaged blood was not used in any of the included studies, and LDF was used in only one study. It was concluded that ICS for cancer surgery was not inferior to allogeneic transfusion [39].

Techniques that have proven to be effective in removing contaminated tumour cells from blood recovered would mainly include LDF and irradiation or photochemical treatment for inactivation [40–43] (Table 2).

**Table 2.** Techniques to Reduce Tumor Load with Metastatic Potential by Using ICS [40–43].

- Optimize oncological treatment prior to surgery: chemotherapy, radiotherapy, immunotherapy;
- Surgical technique supervised by an expert;
- Use of device with effective washing system;
- Targeted suction during surgery: avoid direct contact with neoplasm;
- Extensive wound irrigation and suction to waste when manipulating the tumour;
- Discard blood that may originate directly from a bleeding tumour;
- Use of single or double LDF in-between recovered blood and reinfusion to the patient. These filters are added to the usual system of filtering and transfusion of blood to the patient;
- Irradiation or photochemical treatment of recovered blood for inactivation of tumoral cells.

It has been suggested that the administration of ICS blood through LDF may cause hypotension due to the release of bradykinin when factor XII and platelets are activated by the negatively charged surface of the filters. However, bradykinin and cysteinyl leukotrienes were measured in cell-salvaged blood sampled before and after passage through a negatively charged LDF in 24 consecutive patients with gynaecological or bowel cancer undergoing elective surgery, and no increase in bradykinin concentration was observed after filter passage; in all but one patient, bradykinin was not detectable after filtration ($p$ = 0.007). The change in cysteinyl leukotriene concentration before and after filter passage was statistically non-significant ($p$ = 0.1) [44]. These findings do not support the suggestion that neither bradykinin nor cysteinyl leukotrienes are generated in saline blood during passage through LDF.

### 5.2. Metastatic Spinal Tumour Surgery

Surgery for metastatic spinal and musculoskeletal tumours are associated with significant blood loss and the need for allogenic transfusion. Cell salvage would be a viable alternative, but there is insufficient evidence to generalize its use. Most of the studies conducted are observational. One retrospective controlled study evaluated the safety, efficacy, and cost-effectiveness of ICS in metastatic spine tumour surgery [45]. Overall, there were 63 cases using LDF compared with 113 controls. The results of this study showed that the group with LDF had received lower allogenic blood transfusions (OR = 0.407, $p$ = 0.03), the costs were neutral ($p$ = 0.88) and the average length of stay in hospital was shorter by 3.76 days ($p$ = 0.03). The survival and complication rates between both groups were comparable, with a satisfactory postoperative haemoglobin level.

In addition, several systematic reviews have investigated the use of ICS in metastatic spinal tumour surgery. One provides a weak conclusion that the strategy may be safe in

these surgeries [46]. An extension of this review concluded clinical safety of ICS for this indication, providing evidence for decreased risk for postoperative infections and tumour recurrence, and supported the use of ICS even without LDF in metastatic spinal tumour surgery. This is still controversial for many clinical practitioners [47].

*5.3. Gastrointestinal and Urogenital Cancer*

The question of whether autologous blood transfusion (ABT) during liver transplantation in cases of hepatocellular carcinoma or partial liver resection for metastases from colorectal cancer increases the risk of local recurrence and whether this may have an impact on survival has been addressed by several studies but with inconclusive results.

A retrospective analysis of 147 patients evaluated the impact of ABT on survival and recurrence in patients undergoing partial hepatectomy for liver metastases in colorectal cancer. Seventy-four patients received ICS blood and showed greater intraoperative blood loss, larger resections and longer operation times as compared to those not transfused with ICS. The mean follow-up was 54 months. No difference in survival was found (adjusted hazard ratio (aHR) 0.58; 95% CI 0.31–1.11; $p = 0.10$) and the recurrence-free survival was similar, with an aHR of 0.95 (95% CI 0.54–1.65; $p = 0.85$) [48]. Two other retrospective cohorts were evaluated for the recurrence of cellular hepatocarcinoma and metastases, but there was no difference, and the technique was thus deemed safe. The first cohort included 158 patients; 122 (77.2%) had undergone ICS and 36 (22.8%) were without ICS. No difference in overall or recurrence-free survival was found; neither was there observed a difference in patients with respect to volume infused. The degree of tumour differentiation was the only independent predictor of blood transfusion [49]. In the second cohort of 156 patients, 122 had undergone ICS. No difference was found for medium- and long-term overall survival and for disease-free survival. The ABT was not associated with worse outcomes [50,51]. Concrete reasons to contraindicate ICS in this type of surgery have not been put forward.

An evaluation as to whether ICS may affect short-term survival in open partial nephrectomy for malignant tumours [52] and recurrences, metastatic potential, or mortality in radical prostatectomy [53] has also been attempted through observational studies. However, the results were either non-conclusive or non-applicable to provide general recommendations.

In conclusion, it may be stated that the current evidence does not contraindicate the use of ICS in the field of oncologic surgery, but a general recommendation cannot be established. Specific procedures are recommended to alleviate or reduce tumour load with theoretically metastatic potential, as contaminated cells may be retrieved and re-transfused. These measures include the use of LDF, cell irradiation and exhaustive washing during the surgery (Table 2).

## 6. The Role of Cell Salvage in Peripartum Haemorrhage (PPH)

*6.1. Major Concerns in Obstetrics*

The leading cause of maternal death during childbirth remains uncontrollable haemorrhage. The overall frequency of severe PPH with subsequent hysterectomy has decreased substantially in recent years due to improved uterotonic and damage control techniques, especially in developed countries [8]. The use of ICS in obstetrics may reduce ABT in massive obstetric haemorrhage. In general, the affected population is usually of younger age with potential donors in the postpartum period, so allogeneic blood should be restricted. ICS should be considered in the context of any massive obstetric haemorrhage protocol, provided consensus is reached within the multidisciplinary team, all risks and benefits are individually assessed, and cost-effect is balanced [5–7].

Acceptance of ICS in obstetrics has been hampered by concerns about amniotic fluid embolism and maternal alloimmunization [5–7]. Amniotic fluid embolism is an uncommon but catastrophic obstetric complication that can lead to profound coagulopathy and haemorrhage [54,55]. Because amniotic fluid embolism syndrome is rare (1:8000 to 1:30,000 deliveries), definitive studies assessing its risk with ICS are exceedingly difficult.

In addition, this risk may be overestimated when assessing the use of ICS. The removal of residual amniotic fluid by ICS has been demonstrated, but controversy remains as to whether it is necessary to use one or two separate suction devices for the removal of amniotic fluid and blood [54,55]. Maternal alloimmunization may potentially occur, as ICS cannot differentiate between maternal and foetal RBC. However, the rate of foetal cell transfer in the recovered blood is of the same magnitude as during fetomaternal haemorrhage (FMH) and can be treated by anti-D administration [56–58].

### 6.2. Controversial Use of LDF in PPH

The cell salvage/filtration process may effectively remove plasma elements of amniotic fluid but contamination of salvaged maternal blood by bacteria, amniotic fluid and foetal red cells during Caesarean section, elective, urgent or emergency surgery remains one of the greatest risks [59]. Furthermore, in this setting, an evidence-based approach is challenging [60], and an appropriate evaluation of the ability of LDF to remove potentially harmful components from amniotic fluids is necessary since this seems to be effective [61]. It has been demonstrated that LDF used in addition to cell washing may reduce foetal squamous cell concentration to a level comparable to maternal blood following placental separation, and it has been concluded that washing and filtration produced a maternal-like blood product except for foetal Hb contamination [62]. This exposure to foetal Hb represents isoimmunization and can lead to erythroblastosis in subsequent pregnancies. ABO incompatibility generally tends to be less complex when compared to Rh incompatibility.

A recent study compared the efficacy and safety of LDF and micro-aggregate filters in combination with ICS during Caesarean delivery [63]. Four samples were collected: (i) pre-wash, (ii) post-wash, (iii) post-filtration with LDF and (iv) post-filtration with a micro-aggregate filter. Each sample was analysed for amniotic fluid markers of zinc coproporphyrin-1 and sialyl-Tn, for foetal Hb, and the sample underwent pathological examination for leucocytes and squamous cells. Squamous cells decreased by 59.1% post-wash and 91.2% post-LDF. Leukocytes were removed at a rate of 99.7% by LDF, which was more effective than micro-aggregate filters ($p = 0.02$). Another study [64] reported amniotic fluid markers, described in maternal venous blood at the time of placental separation, which were significantly different from blood recovered from the cell saver post-washout and post-filtration. A certain amount of potassium, lamellar body count and foetal Hb remained after processing.

Leukocyte depletion filters are adhesion filters and allow only slow reinfusion rates. They can become saturated during use, requiring replacement, and have been described to cause bradykinin-mediated hypotension [31,32,44]. The routine uses of LDF and double suction in obstetric practice are currently not recommended. One study investigated changes in blood concentrations of interleukin-1b (IL-1b), interleukin-6 (IL-6), tumour necrosis factor-alpha (TNFa) and bradykinin during passage through ICS and LDF, with or without the application of sub-atmospheric pressure across the filter [65]. Blood samples from 19 healthy women undergoing scheduled Caesarean section showed cytokine and bradykinin concentrations in gravity-filtered blood equal to or significantly lower than those in preoperative venous blood samples. It was concluded that the use of an LDF for gravity-flow ICS blood was likely to be safe. If transfusion of blood using LDF appeared to induce hypotension, an elevation of interleukin-6 could be suspected [65]. Overall, these results remain inconclusive.

### 6.3. Clinical Indications for ICS in Obstetrics

At present, no definitive evidence establishes that ICS in obstetrics reduces ABT or is cost-effective. Most literature originates from case reports, case series and observational studies. There are several perceived barriers and preconceived risks to implementing ICS as a routine procedure in obstetrics since, in addition to the biological difficulties as outlined, there remain concerns related to costs. Strong clinical evidence or economic effectiveness from clinical trials are essential to support a routine practice [66]. If a decision is being

made to use ICS because a patient refuses autologous transfusion or significant blood loss is anticipated, the risk and benefits need to be discussed with the patient and the multidisciplinary team.

Medical indications that have been forwarded for the use of ICS in PPH include severe anaemia, thrombocytopenia, rare blood groups, rejection of ABT and difficulty of blood cross-matching [64]. The more relevant obstetric indications are related to situations of high risk for haemorrhage, like placenta previa, placenta accreta, history of prior uterine rupture, placental abruption, and abnormal placentation. The aim is to discern which patient is most likely to receive a reinfusion of processed blood intraoperatively during delivery or Caesarean section and would be prone to higher blood loss during the procedure.

One study analysed eight years of ICS data from 884 cases of obstetric haemorrhage treated in a regional tertiary care maternity hospital in the UK [67]. In only 21% of cases, enough blood was collected to permit reinfusion. The mean units of reinfused shed blood were 1.2 ± 1.1. Although ICS was most often performed in patients who underwent routine Caesarean delivery (748/884 patients), only 13% of these patients received an ICS reinfusion. Overall, 73% of patients undergoing Caesarean hysterectomy, 69% of those who bled after Caesarean delivery, and 53% who bled after vaginal delivery received an ICS reinfusion ($p = 0.001$) [67].

Targeted ICS in Caesarean deliveries at risk for haemorrhage may be associated with less exposure to allogeneic blood in the operating theatre, but not during the postoperative period. A large observational study from China analysed 11,322 patients pre-implementation of ICS, in comparison to 17,456 post-implementation using an interrupted time series analysis. In the post-implementation period, patients suspected to be at increased risk for the need of blood transfusion (1604, 9.2%) underwent ICS collection. Primary outcomes were the monthly rate of ABT and the incidence of acute blood transfusion reactions. The monthly rate of ABT was lower after implementation (difference −0.7%, 95% CI, −0.1% to −1.4%; $p = 0.03$); however, postpartum, this rate remained unchanged (difference −0.2%, 95% CI, −0.4% to 0.7%; $p = 0.56$). Likewise, the rate of transfusion reactions was unchanged (difference −2%, 99% CI, −9% to 5%; $p = 0.55$) between the two observation periods. The lack of adverse events supported the safety of the procedure [68].

Another retrospective analysis of 115 women evaluated the effect of routine use of ICS on ABT for patients with an anticipated diagnosis of placenta accreta with an estimated blood loss of more or less than 3000 mL, assigned to two subgroups for the revision. The results showed that with ICS, there were less intravenous fluids needed intraoperatively (crystalloids ($p < 0.01$) and colloids ($p < 0.01$)), postoperative length of stay was shorter ($p < 0.01$), and the overall incidence of ABT was lower (OR, 0.179; 95% CI 0.098–0.328). No complications or adverse reactions were observed, and there was no difference reported in the additional use of haemostatic agents [69].

### 6.4. Safety and Cost-Effectiveness of ICS in PPH

An observational study attempted to demonstrate the safety of ICS in obstetric haemorrhage in 1170 patients in whom ICS was routinely used during Caesarean section in the context of the implementation of a PBM program with integrated cell salvage for improving PPH management over a decade [55]. The concept was first introduced in elective cases and then extended to emergencies. As a result, the procedure was effective and safe, and the amount of reinfused autologous blood was, on average, the equivalent of about one unit of allogeneic blood per patient. The median (IQR (range)) volume was 231 (154–306 (80–1690)) mL. There was one suction device used with swab wash and no LDF. It was concluded that the risk of amniotic fluid embolism and foetal red cell contamination should not be considered barriers to the implementation of ICS. During the study, a marked reduction in ABT was noted [55].

Previously, several other studies have attempted to prove the safety of ICS, but with a focus on financial considerations. In a 5-year retrospective review of cases in which ICS was used (587 times) and blood reinfused in 137 patients, the total volume of blood returned

was equivalent to 189 units of RBC. The return rate was higher in urgent than in elective cases (*p* = 0.03). The volume of blood returned through ICS corresponded significantly to estimated blood loss (*p* < 0.00001). Total ICS costs were equivalent to 83 units of blood. It was concluded that routine use of ICS was an appropriate expenditure to reduce ABT, associated with more salvaged blood being returned to patients, which offset the cost of collection sets [70].

To date, the study with the highest methodological quality is the UK SALVO study, but it is without encouraging results [71]. This was a randomized controlled trial at 26 obstetric units involving 3.054 patients during 3 years of routine ICS use, which was the intervention vs. current standard of care as the control group in Caesarean section with a high risk of haemorrhage. The main outcomes were defined with the rate of ABT as the primary outcome, and units of donor blood transfused, time to mobilization, length of hospitalization and FMH were secondary outcomes. Of 3028 patients, 1672 had emergency and 1356 had elective Caesarean sections. The rate of ABT was 3.5% in the control group vs. 2.5% in the intervention group (aOR 0.65, 95%, CI 0.42 to 1.01, *p* = 0.056; adjusted risk difference −1.03, 95% CI −2.13 to 0.06). In a subgroup analysis, the transfusion rate was 4.6% in the control vs. 3.0% in the intervention group among emergency Caesarean sections (aOR 0.58, 95% CI 0.34 to 0.99), whereas this rate was 2.2% vs. 1.8% among elective Caesarean sections (aOR 0.83, 95% CI 0.38 to 1.83, *p* = 0.46). There was no case of amniotic fluid embolism observed. The rate of FMH was higher in the intervention group (25.6% vs. 10.5%, aOR 5.63, 95% CI 1.43 to 22.14, *p* = 0.013), which emphasizes the need for anti-D prophylaxis. LDF was used in 54.9% of cases in the intervention group, with acute hypotension reported [71]. Health Technologies Assessment, based on the results of the SALVO study, made an economic evaluation of the cost-effectiveness of the study, and considered that ICS is unlikely to be considered cost-effective [72,73].

Another study on the cost-effectiveness of ICS was published in the USA [74]. Markov decision analysis modelling compared the cost-effectiveness of three strategies: (i) ICS for every Caesarean delivery, (ii) ICS for high-risk cases, and (iii) no ICS. Each strategy integrated probabilities of haemorrhage, hysterectomy, transfusion reactions and emergency procedures related to ICS, utilities for quality of life, and costs at the societal level. One-way and Monte Carlo probabilistic sensitivity analyses were performed. A threshold of $100,000 per quality-adjusted life-year gained was used as a cost-effectiveness criterion and showed a more than 85% likelihood for ICS to be favourable, being cost-effective (incremental cost-effectiveness ratio, $34,881 per quality-adjusted life-year gained) in cases with high risk for haemorrhage. Routine cell salvage use for all Caesarean deliveries was not cost-effective, costing $415,488 per quality-adjusted life-year gained [74].

In summary, the use of ICS for cases at high risk for obstetric haemorrhage is likely not to be economically effective; routine ICS use for all Caesarean deliveries is obviously not, nor does it appear to be ethical. The findings from previous studies can support the development of guidelines on the management of obstetric haemorrhage, but there is still not sufficient evidence to recommend its use as a routine. Therefore, its current use may be individualized according to standardized protocols to secure maximum safety. Well-controlled and randomized clinical trials are needed to demonstrate that this blood-saving strategy may be considered a priority and cornerstone within the implementation of PBM programs. The routine use of LDF also needs further exploration and standardization.

## 7. Cell Salvage in Acute Trauma and Haemorrhagic Shock

### 7.1. The Risk of Blood Re-Transfusion in Trauma Bleeding

Uncontrolled bleeding remains the major cause of preventable death after trauma, and rapid bleeding control along with correction of coagulopathy are key for patient survival [9]. Damage control surgery and resuscitation principles are applied once the patient is admitted to the trauma bay or even prior to admission. Damage control resuscitation aims to limit further blood loss and addresses the coagulopathy of trauma by combining hy-

potensive resuscitation with balanced crystalloids, early airway control, and the ratio-based administration of blood products and other haemostatic agents.

Current concepts promote early goal-directed therapies based upon results from viscoelastic testing devices such as TEG/CLOTPRO/ROTEM. The latter has been associated with less exposure to potentially harmful allogeneic blood products, thereby reducing the risk of transfusion-related secondary injuries [9].

Recovering autologous blood during traumatic haemorrhage for re-transfusion faces the dynamic difficulty of impaired coagulation (e.g., hypothermia, haemodilution, platelet dysfunction, endothelial damage and glycocalyx degradation, immunomodulation, and proinflammatory effects), and requires a physical cavity to recover the blood, which sometimes comes from multiple points of injury. Therefore, knowing how, when, and where to collect blood during a dynamic multimodal resuscitation strategy is crucial. The control of the bleeding sites may need damage control surgical procedures like thoracotomy or laparotomy.

To date, no official European Quality Standards for the collection and re-transfusion of autologous blood during acute trauma haemorrhage have been established, and the current version of the European Trauma Guideline [9] does not yet mention ICS as a therapeutic approach, although it is practiced in selected trauma centres to reduce and/or avoid allogenic blood products.

The unexpected condition of acute trauma haemorrhage and shock in its most dramatic and dynamic phenotype may hamper the disposal of a cell saver [75]. Blood lost due to mechanical causes and/or disturbed haemostasis may be reinfused if volume and speed of processing along with safety parameters are reached [76]. Blood for re-transfusion must be easily recoverable from body cavities without contamination and should not contribute to further deterioration of haemostasis [7]. If gastric or intestinal perforation is present, the reinfusion of potentially contaminated enteral contents would be contraindicated. Centrifugation and washing during ICS can lead to haemoconcentration and removal of unwanted substances and bacteriological contamination but processing larger volumes could exacerbate the dilution of clotting factors and thrombocytopenia [77]. Some components within recovered blood may participate in favour of oxygen release to damaged tissues, such as 2,3-DPG (3). Nonetheless, residual traces of heparin, micro-aggregates derived from bone chips or fat from fractures and free radicals from haemolyzed erythrocytes [78,79] are non-favourable. In any case, saved blood will be passed through filters of at least 40 microns, with double suction, double filter [80] or with leukoreduction capacity [81]. Cell salvage may be considered an inflammatory trigger through complement activation, interleukin formation [27] and leukocyte activation [30], thereby potentially further contributing to endothelial injury, increased vascular permeability and overall promotion of coagulopathy and haemostatic failure.

At present, there is only a little evidence for the use of ICS in acute bleeding trauma patients [82], although in some combat scenarios studies [83] and selected civilian cases, this approach has been shown to be an effective blood-saving strategy [5,84].

### 7.2. Weak Evidence of Cell Salvage in Civilian Trauma Bleeding

Almost two decades ago, the results from a randomized controlled trial including 44 patients with laparotomy for penetrating abdominal injuries were published on the use of ICS and showed a decrease in ABT within the first 24 h after trauma (control group: 11.17 vs. CS group: 6.47 units, $p = 0.008$) without increasing postoperative infection rates [85]. The mean volume of the reinfused blood retrieved in the ICS group was 1493 mL (range 0–2690 mL). Survival in the ICS group was 7/21 (33.3%) vs. 8/23 (35%) in the control group. Patients who developed postoperative sepsis were more likely to die. However, patients in both groups appeared to be equally likely to develop sepsis, and this was independent of ICS. Overall, 85% of patients in the ICS group had enteric contamination, and 38% had a colonic injury. Of the ICS blood samples sent for culture, more than 90% were positive, with no correlation between the initial microbiologic features of the reinfused

blood and subsequent infectious complications. It was concluded that ICS had no effect on postoperative infection rates or mortality and was associated with a significant reduction in ABT [85]. Since then, no other trial with similar methodology and characteristics has been conducted to either confirm or refute these results [75].

A retrospective matched cohort study of trauma patients treated in a US level 1 trauma centre, reported 47 patients undergoing emergent surgery (83% laparotomies) after trauma who had received ICS and autologous blood transfusions. They were compared to ten similar patients without cell salvage. Patients with ICS had a mean intraoperative blood loss of 1795 mL but received a mean of 819 mL of autologous blood (41% of the transfusion requirement) compared to a blood loss of 978 mL in patients without ICS. Patients with ICS received fewer intraoperative and total units of allogeneic red blood cells compared with no ICS (2 vs. 4 units during surgery ($p = 0.002$) and 4 vs. 8 units ($p < 0.001$ total). They also received fewer total units of plasma (3 vs. 5 units: $p = 0.03$). Mortality was 13% for the ICS group compared with 21% in the no ICS group, which was not statistically different [82].

Whether the bacterial load can be eliminated by washing techniques [85,86] has been debated, as blood contamination with penetrating trauma or enteral injury can lead to disseminated infection. One option to counteract this risk may be the administration of antibiotics at the time of re-transfusion of autologous blood [87].

### 7.3. Combat Injury, Combat Blood

A feasibility study of ICS in an Anglo-American combat support hospital was conducted in 130 patients with combat-related injuries, of which 27 patients required massive transfusion, defined as >10 units of RBC within the first 12 h of injury. In 17 patients, ICS was completed. Of these, the mechanism of injury was, in 24% of cases, a gunshot wound and, in 76% of cases, a blast injury, and the type of injury was, in 47% of cases, a body cavity injury and, in 53% of cases, an extremity (the retrieval of blood from limb injuries was technically difficult and only low volumes were recovered). On average, for all patients, autologous blood accounted for only 7.6% of the total requirement. In patients undergoing more extended cavity surgery (i.e., laparotomy or thoracotomy) following gunshot injury, the ratio of RBC mass recovered to that required was 39% [83].

Another retrospective review was conducted in 2014 on emergent trauma resuscitation in 179 patients with penetrating and blunt abdominal trauma requiring intraoperative blood transfusion [26]. Comparing 2 groups, 1 group received only bank blood ($n = 108$) and the other also received autologous blood ($n = 71$). Multivariate regression analysis was applied to assess primary outcomes, survival, and infection. The results showed no significant difference regarding age, Injury Severity Score (ISS) pattern, length of stay, postoperative International Normalized Ratio (INR) and volume of stored blood product volume. Both groups were also proportional to colon injury. The intraoperative blood loss was $2472 \pm 3261$ mL for controls and $4056 \pm 3825$ mL for the auto transfused group ($p = 0.0001$). The total volume of blood transfused was 2792 and 5513 mL for the controls and the autotransfusion group, respectively ($p = 0.002$). Overall, 90 controls (84%) and 53 patients with ICS (76%) survived until discharge ($p = 0.21$). Logistic regression analysis revealed that ISS >25, systolic blood pressure <90 mmHg and estimated blood loss >2000 mL predicted mortality. There was also a trend for decreasing survival with age >50 years and no evidence that emergent autologous transfusion worsens clinical outcomes [26].

These data suggest that the blood salvaged from body cavities after military trauma could be safely transfused and could significantly reduce the blood bank's supply in selected patient groups.

The current evidence available for abdominal or thoracic trauma remains feeble. Randomized controlled trials are needed to assess efficacy, safety, and cost-effectiveness in the context of severe trauma emergencies, and it would be necessary to elaborate a multidisciplinary protocol.

The cost-effectiveness of ICS in the civilian setting, which includes transporting donor blood, seems to be a different model compared to the military setting, where whole blood can be used. Concerns exist on how effective prehospital salvage and autotransfusion could be under field conditions.

Contemporary military transfusion medicine is focused on haemostatic support, which involves the acute coagulopathy of trauma, damage control resuscitation, protocolized massive transfusion, whole blood, and walking blood banks [88–91]. In combat medicine, the traditional use of ICS is neither practical nor cost-effective, so the possibility of new technologies would be considered, such as the use of a membrane-controlled superabsorber instead of centrifugation, that can remove plasma from the blood. This could offer advantages in a resource-constrained environment but requires further evaluation to determine the feasibility of prehospital salvage after combat injury, the quality of the 'blood component' of the product available for autotransfusion and the impact of any transfused product on both a normal and coagulopathic haemostatic system [92].

### 7.4. Cell Salvage in Specific Injuries
#### 7.4.1. Pelvis Fractures

In acute pelvic trauma with haemorrhage, ICS could be indicated when fracture management involves open anterior surgery. For the initial treatment of unstable pelvic ring fractures with severe bleeding, damage control surgery (DCS) techniques are applied. Rapid closure of the pelvic ring is key to decrease intrapelvic volume and provide counter-pressure against the bleeding source for tamponade. Pelvic binders in the acute setting followed by external fixation are indicated in most cases, where there is no possibility to use an ICS. However, if an open reduction internal fixation approach is performed, for example, with an infraumbilical laparotomy or with preperitoneal pelvic packing, then there is a potential opportunity for ICS. The transfusion of blood products is required in around 10% of all pelvic fractures [93]. Clinical classification of pelvic fractures is important for risk stratification, as the more severe and unstable the injuries are, the higher the likelihood for surgical intervention is [93,94].

Acetabular fractures are usually stable, and they can be managed with an anterior or posterior approach for definitive reduction and fixation. An indicated damage control orthopaedic procedure in these cases is trans-skeletal traction.

Pelvic ring fractures can be unstable and are treated with internal fixation via the anterior approach or with a posterior percutaneous screw. Orthopaedic damage control (ODC) in unstable fractures is external fixation, open surgery with pelvic packing or both.

Pelvic fractures can also be combined, involving the acetabular and the pelvic ring. This often complicates the final therapeutic decision, although the control of severe bleeding in an unstable fracture is always determined by ODC measures: the external fixator, C-clamp (previously the pelvic binder or wrapping with a sheet), radiological arterial embolization and, if preperitoneal pelvic packing is also required, infra-umbilical laparotomy [93,94]. Only in the latter case, ICS would be considered in the acute moment of the bleeding unstable pelvic fracture and as definitive treatment in those procedures in which open surgery is also performed but in deferred surgical times. Only in extremis, the management of an unstable severe bleeding pelvic fracture is managed by a REBOA (resuscitative endovascular balloon occlusion of the aorta), which in any case would be an indication of ICS.

Osteosynthesis of complex fractures of the acetabulum are not emergent surgeries and, due to their technical difficulty, must be deferred.

The results from observational studies on ICS are neither conclusive nor definitive enough to recommend its widespread use. The mean blood volume re-transfused was between 388 mL and 484 mL, and practically every study concluded that the more complex the fractures display, including concomitant injuries and the greater the bleed, the more cost-effective the use of ICS would be, even more economic than allogeneic transfusion [95,96]. One retrospective single-centre cohort of 145 patients identified the anterior approach as a

risk factor for elevated blood loss and ICS return [97]. Another retrospective evaluation could not find any difference in allogeneic transfusion, and the postoperative rate was higher in ISS >20, with no difference for ICS vs. non-CS groups or technical approach [98].

As no routine use for ICS in acute pelvic trauma, including fractures, can yet be recommended at this stage, this procedure may be warranted if extended blood loss is anticipated in open reduction internal fixation (ORIF) of acetabular fractures and when an anterior approach is performed [94].

7.4.2. Haemothorax

Autologous fresh whole blood from the thorax cavity could be recovered and re-transfused in case of chest trauma through the salvage of mediastinal blood from chest drains after traumatic haemothorax or after thoracotomy. In open surgery, ICS would be provided by washing, filtration, and the application of broad-spectrum antibiotics, since the procedure is performed in the context of a potentially contaminated environment in cases of penetrating trauma. The main benefit of ICS would be a reduction in the need for allogeneic blood transfusion. However, ICS has not been widely adopted yet in trauma due to its potentially negative effects on coagulation and inflammation and due to logistical challenges. To date, there are no large, controlled, and prospective studies. Data from observational studies have provided considerable safety, but cost-effectiveness has yet to be shown.

A multi-institutional retrospective study from two Level I trauma centres in the US was conducted with some beneficial results [99]. Overall, 272 trauma patients were allocated to two study groups, i.e., (i) non-autotransfusion (AT) and (ii) autologous whole blood transfusion from the haemothorax. There was no significant difference between groups for in-hospital complications ($p = 0.61$), mortality ($p = 0.51$), and 24-h post-admission coagulation. Patients who had received AT had significantly lower requirements for allogeneic blood ($p = 0.01$) and platelet concentrates ($p = 0.01$). The cost of transfusions ($p = 0.01$) was significantly lower in the AT group [99].

Several experimental studies [100–103] have assessed the paradoxical characteristics of the composition of blood retrieved from a haemothorax. One first question was whether the composition of blood recovered would be equal or similar to fresh whole blood, but it was found to be haemodiluted. The recovered blood could be therefore considered a potential driver or promoter of coagulopathy when analysed independently, but the same blood was rendered hypercoagulable when mixed with plasma. Traumatic haemothorax has been shown to contain predictably low fibrinogen, low haematocrit, and low platelet counts. When analysed separately and in isolation, traumatic haemothorax reflects a coagulopathic environment and state. However, when mixed with normal plasma at physiological dilutions, the haemothorax shows enhanced coagulation. It may be observed that, in vitro, an evacuated haemothorax has the potential to induce a highly active coagulative state. The coagulation status should be then monitored closely when proceeding to an ICS, and viscoelastic tests could be the best choice for assessment and monitoring. According to these experiments, autotransfusion of a haemothorax would likely result in a hypercoagulable state in vivo. However, it may be useful in resource-limited settings [100–103].

*7.5. 'Cross-Talking' Inflammation-Coagulation-Immunity in Trauma*

Allogeneic blood transfusion has been associated with transfusion-related immunomodulation (TRIM) and subsequent poorer outcomes, including perioperative infection, multiple organ failure, and mortality [104]. The transfusion of RBCs alone does not create an immunological response in healthy surgical patients, but in severe trauma, the response may be different [104]. Traumatic injuries have a profound immunomodulatory effect, affecting a wide range of immune components. This may involve the disruption of organic macro- and micro-barriers, thereby inducing the activation of innate immunity [105–107]. These sequalae may further be complicated by transfusion. Blood transfusion is associated with a gene transcription profile of immunosuppression; there has been a reorganization of

the genomic storm described [108,109]. Besides, immunomodulation is associated with leukocyte-depleted blood (exposure to foreign antigens) [110,111]. Cell salvaged blood is fresh, washed and leukocyte-depleted to avoid contamination and theoretically would not mirror all the disadvantages of allogeneic blood, although the actual immunomodulatory potential of autologous re-transfused blood is yet to be identified.

There is also the possibility of transfusion-related acute lung injury (TRALI) with autologous blood re-transfusion, although this is an adverse effect that has been related more to plasma transfusion [112–115]. The SALVO study [71] discussed this complication without concluding a serious clinical impact.

The host response to trauma typically involves a strong immunosuppressive component, which, in contrast to the systemic inflammatory response syndrome/compensatory anti-inflammatory response syndrome model, occurs early and overlaps with proinflammatory and antimicrobial elements [116,117]. Still, following a major inflammatory insult such as after severe trauma, an inflammatory and immunosuppressive response occurs simultaneously. Early death from acute multiorgan failure (MOF) is now rare due to the early recognition of shock followed by rapid supportive care. Survivors can evolve along two pathways: (i) they may easily return to immune homeostasis and make a rapid recovery, or (ii) they remain in the ICU and develop chronic inflammation, suppression of adaptive immunity, ongoing protein catabolism with cachectic wasting, and develop recurrent nosocomial infections [116,117].

In a small prospective observational study, an in vitro model was used to assess immunocompetence through the assessment of intracellular cytokine production [118]. Co-stimulation and adhesion molecules expressed on dendritic cells and on monocytes and the overall general immune responses following allogeneic blood transfusion or ICS exposure were measured. The exposure to both allogeneic blood transfusion and ICS suppressed dendritic cell and monocyte function, but suppression was significantly less following ICS. It was concluded that ICS presented an improved immune competence. This modulation of the overall leukocyte response may predict a reduction of adverse outcomes following ICS, for example, infection [118]. According to this theory, the use of ICS in severe traumatic procedures would be more immunologically beneficial than allogeneic transfusion.

*7.6. Overall Considerations for ICS in Trauma Haemorrhage*

Cell salvage may hold the potential to reduce allogeneic blood product transfusion in severe trauma haemorrhage, but the evidence remains weak. There is still concern for worsening coagulopathy and inflammatory disbalance, and when haemorrhagic shock occurs, it fights against a complex and highly dynamic environment. Patients should be evaluated with Trauma Scores and type of injury, e.g., pelvic bleeding and haemothorax. The main resources to prevent and avoid contamination are washing, LDF, and different alternatives of centrifugation. Nanotechnology could help with the development of special filters and alternative concentrator bags or other devices that could separate and re-transfuse platelets. Military settings must be clearly distinguished from the civilian setting and the urban hospital. It must be recognized that ICS may also be considered a lifesaving option for those patients who refuse transfusion. In summary, ICS is not yet cost-effective at this stage but may become a valid option if selected blood loss thresholds are defined, a non-contaminated cavity for collection is present, and good logistics and infrastructures are secured to select patients more judiciously.

## 8. Measures to Improve ICS Implementation in PPH and Trauma

Several improvement measures may be suggested (Table 3).

**Table 3.** Roadmap to improve the implementation of ICS in obstetrics and acute trauma bleeding.

| | |
|---|---|
| ✓ | Strategies comparison analysis for cost-effectiveness: accurate selection of patients: not for all trauma bleedings, not for all Caesarean sections |

**Table 3.** *Cont.*

| | |
|---|---|
| ✓ | Select which high-risk haemorrhage could be economically reasonable: probability of haemorrhage >30% estimated blood volume<br>Trauma in thoracotomy, pelvic packing or open fixation, laparotomy, emergency procedures in a cavity, non-contaminated preferred<br>Obstetrics: high risk bleeding, placenta accreta |
| ✓ | High-speed washing devices with optional LDF (if contamination), cell concentrator bags |
| ✓ | ICS not indicated if severe shock or coagulopathy from the beginning |
| ✓<br>✓ | Evaluated (evolution) decision—use when is possible<br>Debriefing with the trained multidisciplinary team |
| ✓ | Technical preparation and logistics (senior medical advisor) |
| ✓ | Trained team with predefined instructions in debriefings—nursing roles |
| ✓ | Define the type of trauma injuries: vascular lesion (thorax, pelvic or abdomen), solid organ lesion (liver, spleen, cardiac, lung), pelvic ring fracture |
| ✓ | Not feasible in mass casualties, concept of combat setting and whole blood resources: cell concentrator bags |
| ✓ | Option when Jehovah's Witnesses |
| ✓ | Integrate ICS in severe bleeding protocols, in the hospital haemovigilance programme, blood bank and Hospital Transfusion Committee |
| ✓ | Quality assurance manual, regular quality controls, review of results |

## 9. Conclusions

The scientific evidence and experience related to ICS in oncological surgery, PPH and trauma remain a topic of ongoing discussion and controversy. In some locations and settings, ICS is used on a case-by-case decision with safety measures in place and apparently good results, but without demonstrating its cost-effectiveness. No study has yet been conducted to either promote or refute its use, to allow generalization and to consent to firm recommendations. The use of ICS requires continuous updating and education [119], especially in the surgical specialties discussed in this work. There is a window of opportunity, and high-quality studies are urgently needed.

**Author Contributions:** Conceptualization, L.M.M.; literature search, L.M.M.; writing—original draft preparation, L.M.M.; writing—review and editing, L.M.M., M.M. and S.M.M. All authors have read and agreed to the published version of the manuscript.

**Funding:** This research received no external funding.

**Institutional Review Board Statement:** Not applicable.

**Informed Consent Statement:** Not applicable.

**Data Availability Statement:** Not applicable.

**Acknowledgments:** This study was performed within the framework of a Doctorate Degree in Medicine in the Autonomous University of Barcelona, Barcelona 08035 Spain (L.M.M.). All authors consented to this acknowledgment.

**Conflicts of Interest:** The authors declare no conflict of interest.

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
