# Peer review of "Cell Salvage in Oncological Surgery, Peripartum Haemorrhage and Trauma"

_2673-4095, doi:10.3390/surgeries3010007_

Round 1
Reviewer 1 Report
The current version of rhe manuscript has been greatly improved in respect of the previous submission and is now suitable for publication.
Author Response
Thank you very much for the new work of correction and evaluation of the manuscript and I do appreciate your consideration that the text is now suitable for publication.
Reviewer 2 Report
The author’s submitted revised version of a narrative review in regards to cell salvage in different conditions such as oncological surgery, peripartum bleeding and trauma patients.
The authors made an effort and revise the manuscript but unfortunately I am still not happy with this text and they did not respond adequately to any of my comments:
- Even this is narrative the aims and investigated outcomes should be pointed. I do not see that the authors add any of these.
- Although this is narrative review methodology and literature review strategy should be clearly stated. Date when the literature search was performed ad who performed literature review (initials) should be stated as well. As I pointed previously minimum of 4 databases search should be performed. (REFERENCE: Bramer WM, Rethlefsen ML, Kleijnen J, Franco OH. Optimal database combinations for literature searches in systematic reviews: a prospective exploratory study Syst Rev. 2017, 6, 245.)
- Most of the chapters are still very lengthy, poorly structured, and not easy to follow (like a chapters from a book).
In conclusion, this is just repeating facts from the literature without any comparison. I do not see any significant improvement. I really do not see any benefits for the readers from this text.
Author Response
Dear reviewer, thank you very much again for your critical reading of the paper after having made the suggested changes. We will again try to include some needed improvements.
Regarding the methodology, as it is not a systematic review, the search procedure and the bibliographic databases used have been included and the main objective of the review, but there is no detailed description of the literature search, as it would correspond to a systematic review or a meta-analysis. The brief description of the methodology is included in the introduction of the article, in the first paragraphs. However, we try to write more precisely according to your indications, and we also state at the end of the paper who conducted the search (LMM). Your recommended article is very appreciated.
Thank you very much for pointing out that the length of the text can be excessive and therefore difficult to understand. In previous changes, we had tried hard to improve this tendency and introduce dynamism, however, it is maybe still too dense. We will try to free some superfluous content from the text so that the new version to be sent will be more agile. We realise that the author is sometimes unaware of the impact of such a long text, as he or she is writing and does not have an external or broader vision.
We really appreciate your sincere impression that the text appears to be a redaction of previous published studies with no intention to compare or evaluate significant changes. Nevertheless, the intention of this work was to compile and summarise the evidence so far on a topic that is controversial in precisely these three clinical settings. Unfortunately, there is not much evidence and therefore, there have been no major changes in recent years. It would be inappropriate for us to openly express our opinion on the use of ICS in these three pathologies, but the description of previous clinical experience as reported in the literature on the subject is quite optimistic in some cases but reflects a low cost-effectiveness. Therefore, the reader should be able to establish his or her own criteria based on this review, which above all aims to objectify the scarce generalised experience in its recommendations, neither in the clinical guidelines nor in the literature.
We would like the benefit for the reader to be precisely that critical view that this is a very controversial subject, difficult to reach consensus on, and therefore there is not enough scientific evidence and generalisations in international guidelines.
Following your suggestions and that of the other reviewers, we will specifically try to reduce redundant content to make it easier to read and thus to understand, and hopefully the reader will be able to elaborate such a critical opinion. Thank you very much again for your effort. We really appreciate your evaluation.
The major changes will be done in the word version, that hopefully will be submitted after the evaluation of the responses to the comments.
Reviewer 3 Report
Mora Miquel et al. resubmitted their previously rejected narrative review entitled ‘’Cell Salvage in Oncological Surgery, Peripartum Bleeding and Trauma Patients’’.
After careful reading of the new text, as well as the author’s answers to my previous comments I have to conclude that not much has changed in this manuscript. Although this was narrative review the authors should significantly improve methodology: they should clearly state included studies, methods of search, outcomes, flow chart. I do not see any of that in revised version of the manuscript. Also, as I pointed previously there is no clear aims, as well the outcomes that were investigated.
This text is very extensive, poor in structure and repeating the well known facts, with no clear synthesis of presented results. The conclusions presented in different paragraphs are not new.
En explanations received from the authors in regards to my previous comments are premature, as they stated that my comment in regards to text plagiarism were due to the lack of a refined style… I did not receive wrong impression that a text was plagiarized, I found copied sentences from previous publications! Moreover, style is generally very poor and not easy to follow.
Finally, this text is of very low significance. The authors were previously advised to perform systematic review which has significantly higher value but they just adopted previous text without significant changes.
Author Response
Dear reviewer, thank you very much again for your critical reading of the paper. We will try to include more improvements, to ameliorate your disappointment with the changes that were made. We are sorry that they did not meet your expectations following your suggestions, which we will try again to follow up carefully.
Regarding the methodology, as it is not a systematic review, the search procedure and the bibliographic databases used have been included and the main objective of the review, but there is no detailed description of the literature search, methods of search, outcomes, and a flow chart as it would correspond to a systematic review or a meta-analysis. The brief description of the methodology is included in the introduction of the article, in the first paragraphs. However, we try to write more precisely according to your indications. We also included who performed the literature search at the end of the article.
The clear aim of this review intended to provide a descriptive but also critical assessment of the use of ICS in three clinical scenarios where there is considerable controversy. An attempt has been made to describe this objective in the introduction. Specific outcomes as if we had conducted a clinical study are not possible. We have tried to add in each section and at the end, a short conclusion summarising the previous clinical experience referred to in the literature.
Thank you very much for pointing out that the length of the text can be excessive and therefore difficult to understand. In previous changes, we had tried hard to improve this tendency and introduce dynamism, however, it is probably still too dense. We will try to free some superfluous content from the text so that the new version to be sent will be more agile, above all, as you indicate, we will try to synthesise the results of each section. Some of these partial conclusions have been included to some extent, as mentioned, but we note that they are not sufficient.
We thank you again for your critical view of the style. In our previous explanations we tried to explain why some of the expressions appear to be identical to those in other publications. Sometimes, the transcription of the results ends up being a possible copy and the translation of expressions that in different native languages, when translated into English, can result in artificial translations or in a style that is less appropriate for a scientific publication. We try to resolve these incidents by reviewing the entire text and again we apologise if we have given the impression of plagiarism or if some phrases are very similar to what was previously written. The idea of writing this review came because of a presentation on exactly this topic at an international meeting, so the main authorship of the text has moderated the composition of the whole text and its content. We are sorry for the inconvenience caused by the style, and we will try again to improve it to make it suitable for a new evaluation on your part.
We thank you for your advice that the text should be converted into a systematic review, which I absolutely agree is of much higher scientific quality. However, taking up the narrative review work that had been presented in a panel discussion, and after expanding the literature search, we thought it was a good option to record the controversy of this topic, which unfortunately does not have much previous scientific evidence. The authors of this text received an invitation from an editor of the journal to choose a free format and a topic related to PBM, also of our free choice. We believe that these are three topics on which no international consensus has been reached and therefore a recommendation in clinical guidelines cannot be generalised given the lack of methodologically comprehensive high-quality studies.
If the editor and you consider it appropriate, we will send a version with improvements for further evaluation. Thank you again for your time and interest. It is very appreciated.
Reviewer 4 Report
Dear authors,
thank you very much for preparing the present paper.
The topic is interesting and certainly important for perioperative medicine.
However, I have some major doubts about the approach and design of the manuscript. Firstly, why did you choose for a combination of three different situations in which cell salvage might be used. Obstetric bleeding might be totally different from trauma and from cancer surgery. Each topic might fill a review on its own. The intensive elaboration on different cancers is an example for the broad character of this part of the review.
In addition, the manuscript seems built up strangely. Maybe you should first introduce the technique of CS, then mention some common areas of use, before you start with the detailed discussions.
Table 1 seems quite vague. Yes, these are agents which should not be infused, but the question in how far CS-blood is cleared from all of this should be described, too.
Although in modern medicine important, I doubt that the repeated argument of cost-effectiveness overrules potentials benefits of CS and alleviates unwanted effects of RBC transfusion.
My recommendation:
-shorten the text
-give it a different structure
-maybe consider focussing on one bleeding situation instead of three.
Kindly

Author Response
Dear reviewer, I am very grateful for your meticulous reading of the review and for all your comments, which I have tried to answer in the most concrete way to dispel the doubts that the text may create. Also, and following all your indications, I have been making changes in the text in word format, which will be seen in the version that I will resubmit, so that it can be re-evaluated. Once again, thank you very much. Please see the attachment.

Round 2
Reviewer 2 Report
As I pointed out several times previously, this paper is a very lengthy, poorly structured narrative review which does not have any value in scientific literature as this is mostly just repetition of well-known facts. All of this is available in current medical literature. Unfortunately, I do not see much benefits for the readers from this review. In the revised version I do not see any significant improvement. If the authors are willing to perform systematic review (or even more) meta-analysis this text may be reconsidered. In this form this is not suitable for publication.
Reviewer 3 Report
This is fourth time that I recieve this manuscript for evaluation and nothing significantly changed.
This text is very extensive, poor in structure and repeating the well known facts, with no clear synthesis of presented results. The conclusions presented in different paragraphs are not new.
En explanations received from the authors in regards to my previous comments are premature, as they stated that my comment in regards to text plagiarism were due to the lack of a refined style… I did not receive wrong impression that a text was plagiarized, I found copied sentences from previous publications! Moreover, style is generally very poor and not easy to follow.
Finally, this text is of very low significance. The authors were previously advised to perform systematic review which has significantly higher value but they just adopted previous text without significant.
This manuscript is a resubmission of an earlier submission. The following is a list of the peer review reports and author responses from that submission.
Round 1
Reviewer 1 Report
The authors reviewed literature about an important issue in medicine.
The study is well organized and comprehensive, well written and informative.
I have one suggestion. The authors should discuss about the risk of Transfusion-related Acute Lung Injury (TRALI) related to cell salvage procedure, in particular in autologous transfusions procedures. However, this issue was faced in the cited SALVO study.
Reviewer 2 Report
The authors performed a narrative review in regards to cell salvage in different conditions such as oncological surgery, peripartum bleeding and trauma patients.
I read the study with interest. Unfortunately this study lacks in fundamental issues as follow:
- There is no clear aim of the study. Main text is very wordy without clear aims. Also, it is unclear what the primary and secondary outcomes of this review were. What exactly did the authors exactly want to investigate?
- The main weakness of this review is that there is no methodology presented. What parameters / variables were included in analysis, there is no search strategy presented. This should be clearly stated in methodology. How the authors identified studies for this so-called narrative review. Also there is no synthesis of the results from the included studies.
- Which databases were used for search of the literature? At least four databases need to be explored for an efficient search in reviews. (REFERENCE: Bramer WM, Rethlefsen ML, Kleijnen J, Franco OH. Optimal database combinations for literature searches in systematic reviews: a prospective exploratory study Syst Rev. 2017, 6, 245.) Even though this was a narrative review the authors should present fundal methodology.
- The main weakness of this study is that there are no results. None variable has been studied and there is no presentation of the results. This narrative review looks like a chapter from the book. All of this is well known and reported several times in medical literature.
- Most of the chapters are very lengthy, poorly structured, and not easy to follow. The authors do not focus on any main outcome (actually main outcomes are unclear). This is just repeating facts from the literature without any comparison.
Unfortunately, I do not see much benefits for the readers from this review. This is not enough for publication in international journal.
Reviewer 3 Report
Mora Miquel et al. presented narrative review entitled ‘’Cell Salvage in Oncological Surgery, Peripartum Bleeding and Trauma Patients’’.Despite very lengthy and boring text which is mostly copied from published literature the main objection of this narrative review is very poor design. There is no methodology section, also there is no any results. Moreover, I do not see flow chart and the results of searched databases, which data bases were evaluated? How many articles were identified, which were analyzed, which keywords were used, how many articles were identified and analyzed, what were inclusion and exclusion criteria. Actually it is totally unknown what the purpose of the study was. No clear aims were identified by the authors. As we can conclude we do not know nothing about study design which is one of the most important parts of reviews. The authors should seek help how to perform review of literature. Also, I would advise the authors to perform systematic review (or even meta-analysis), this narrative reviews are of low interest to the readers and are the lowest level of evidence. Any serious journal should not even consider reports like this for review. Conclusions are well known. This review does not represent any novelty. Many sentences has just been copied from existing articles, we can see it from copied values from studies… e.g. the authors copied p values ‘’<.0001’’ in other part of the text they used different style of presentation ‘’p<0.0001’’. The article should be checked for plagiarism.